# Help seeking behavior by women experiencing intimate partner violence in india: A machine learning approach to identifying risk factors

**Nabamallika Dehingia**[1,2]*, **Arnab K. Dey**[1,2], **Lotus McDougal**[1], **Julian McAuley**[3], **Abhishek Singh**[4], **Anita Raj**[1]

**1** Center on Gender Equity and Health, Department of Medicine, University of California San Diego, San Diego, California, United States of America, **2** Joint Doctoral Program-Public Health, San Diego State University and University of California San Diego, San Diego, California, United States of America, **3** Department of Computer Science, School of Engineering, University of California San Diego, San Diego, California, United States of America, **4** International Institute of Population Sciences, Mumbai, India

* ndehingi@ucsd.edu

**Data Availability Statement:** This study used Indian National Family Health Survey-4 (2015-16) dataset, which is available in public domain. The

## Abstract

### Background

Despite the low prevalence of help-seeking behavior among victims of intimate partner violence (IPV) in India, quantitative evidence on risk factors, is limited. We use a previously validated exploratory approach, to examine correlates of help-seeking from anyone (e.g. family, friends, police, doctor etc.), as well as help-seeking from any formal sources.

### Methods

We used data from a nationally-representative health survey conducted in 2015–16 in India, and included all variables in the dataset (~6000 variables) as independent variables. Two machine learning (ML) models were used- L-1, and L-2 regularized logistic regression models. The results from these models were qualitatively coded by researchers to identify broad themes associated with help-seeking behavior. This process of implementing ML models followed by qualitative coding was repeated until pre-specified criteria were met.

### Results

Identified themes associated with help-seeking behavior included experience of injury from violence, husband's controlling behavior, husband's consumption of alcohol, and being currently separated from husband. Themes related to women's access to social and economic resources, such as women's employment, and receipt of maternal and reproductive health services were also noted to be related factors. We observed similarity in correlates for seeking help from anyone, vs from formal sources, with a greater focus on women being separated for help-seeking from formal sources.

dataset can be accessed from the Demographic Health Survey(DHS) website: https://dhsprogram. com/data/dataset/India_Standard-DHS_2015.cfm? flag=0.

**Funding:** This study was funded under a grant from the Bill and Melinda Gates Foundation (Grant number OPP1179208; PI: Anita Raj). The funders had no role in study design, data collection and analysis, decision to publish, or preparation of the manuscript.

**Competing interests:** The authors have declared that no competing interests exist.

## Conclusion

Findings highlight the need for community programs to reach out to women trapped in abusive relationships, as well as the importance of women's social and economic connectedness; future work should consider holistic interventions that integrate IPV screening and support services with women's health related services.

## Introduction

Despite the significant global attention received by intimate partner violence (IPV) prevention efforts in the past two decades, IPV continues to be a pervasive social problem, across geographies [1–3]. Experiences of IPV can impact many aspects of women's well-being, including social cohesion and connectedness, economic security, physical and mental health, and political aspirations [3]. There is evidence that IPV has increased under the COVID-19 pandemic [4], and possibly more so in contexts with higher COVID-19 prevalence such as India. Most recent evidence from India, prior to pandemic, demonstrates that one in every three married women has experienced physical, and/or sexual spousal violence at least once in their lifetime [5]. These figures are likely underestimations, given the stigma around gender-based violence victimization [6]. Nonetheless, these findings indicate that at least 86 million women in India have experienced physical and/or sexual violence at the hands of their husband [7]. This violence is reinforced by pervasive attitudes of acceptance and justification of IPV in the country [5] as well as limited availability of local support services for victims [8, 9]. Unsurprisingly, help-seeking among those affected by violence remains low in the country.

Among women in India who have ever experienced physical or sexual violence, only 14% reported formal or informal help-seeking, with formal help-seeking far less likely than informal help-seeking (e.g., 65% of help seekers turned to family where <5% of help seekers turned to police, social services, or health services for support) [5]. These latter findings are similar to that seen across a number of other country contexts [10]. Further, evidence from 2006–07 to 2015–16 in India indicates a decline in women's help seeking. Given the demonstrated importance and value of disclosure and support services for victims of IPV in India [9, 11], we need greater understanding of what factors are associated with IPV help-seeking, with the goal of increasing this behavior. In this study, we aim to identify potential correlates of help-seeking behavior by victims of IPV in India, using an exploratory approach and machine learning models. This hypothesis-generating analysis offers a means of highlighting factors related to help-seeking in a context of high IPV prevalence and low help-seeking.

Existing literature highlights several individual, societal, and legal barriers to women's help-seeking behavior and/or disclosure of IPV experiences [12, 13]. Research from high- income countries has noted low educational status, unemployment, and poor economic status as factors associated with women choosing not to seek help, and remaining in abusive relationships [14, 15]. In the United States, cultural prescriptions against seeking help prevent women belonging to ethnic minority groups including Hispanic women, from reaching out to legal or formal support services [16]. In situations where IPV victims do not have economic independence, worries about child support and economic survival can also act as a barrier to seeking help [17]. Studies from South Asia, including India, have emphasized the key role played by existing patriarchal norms around marital relationships on IPV perpetration as well as help-seeking. Fear of social repercussions, fear of jeopardizing family's honor, and fear of divorce often prevent women from seeking help [18–20]. Absence of strong legal institutions with a

mandate to protect women from gender-based violence can also discourage women from seeking help, with qualitative findings from India suggesting that in many cases, police dismiss cases of IPV as a 'private matter' between the husband and wife (18). In contrast, factors increasing the likelihood of help-seeking in certain settings in Sweden and New Zealand include experiencing psychological distress, and having children with the perpetrator [21, 22]. With the exception of qualitative studies with specific groups of women, research on correlates of women's help-seeking behavior for IPV in India, is limited [8]. The lack of quantitative evidence may in part be due to the low prevalence of help-seeking behavior, which can create challenges with regards to implementation of traditional statistical models.

With this study, we aim to fill this gap in literature, and identify potential factors associated with women's help-seeking behavior for IPV in India. We examine correlates from a large group of variables related to women's socio-demographics, health outcomes, agency, and experience of violence, using an exploratory approach previously validated in India [23, 24]. This technique uses machine learning regression models that allow us to address the issues associated with examining low-prevalence outcomes and large number of independent variables. This approach also allows for an exploratory lens of an analysis rather than an a priori hypothesis driven approach. With recognition of the unique distinctions between disclosure and help-seeking broadly, and help-seeking with more formal institutions, we include both forms of help-seeking as outcomes, allowing for hypothesis generation for testing via future work and guidance toward potentially new targets for help-seeking interventions.

## Materials and methods

### Data

Data used for the study was obtained from the fourth round of India's Demographic and Health Survey (DHS) conducted in 2015–2016 [25]. The survey covered a nationally representative sample of women in the age range of 15–49 years, and included a wide range of questions on socio-demographic characteristics of women, sexual and reproductive health, fertility history, maternal and child health, access to health services, and women's agency and empowerment. The survey also included questions related to women's experiences of violence, administered to a sub-sample of women. This study includes this sub-sample of women who are or were married, and who reported to have experienced physical and/or sexual violence perpetrated by their spouse, at least once in their lifetime (N = 19,468). Experience of physical and/or sexual violence was measured through a list of standard questions, used by the demographic health surveys across different countries. The current study does not cover emotional violence as an outcome, given that emotional IPV is often not as agreed upon as physical and sexual IPV, as indicative of abuse and hence requiring help-seeking. Our study focusses on help-seeking for women experiencing sexual and/or physical IPV- the forms of violence that are recognized more consistently by service organizations, the criminal justice system, as well the society.

### Measures

**Dependent variable.** Our analysis examined two outcome variables: a) IPV help-seeking from anyone (formal institutions and/or family and friends), and b) IPV help-seeking from formal institutions. The first outcome variable (Yes/No) was measured based on response to the question- "*Thinking about what you yourself have experienced among the different things we have been talking about, have you ever tried to seek help*?". Those who responded with a 'Yes' to this question were then asked about who they sought help from. Women who reported to have

sought help from the police, a lawyer, a doctor, or a social service organization were categorized as 'IPV help-seeking from formal institutions'.

**Independent variables.** We adopted an exploratory approach to identify variables associated with IPV help-seeking. Hence, the complete DHS dataset, with limited exceptions, was included as potential independent variables. Two researchers (ND, AD) in the team reviewed each variable in the dataset to identify repetitions and redundant variables, as well as survey design and structure variables, which were excluded from the analysis. For example, variables related to date of interview, respondent IDs etc. were removed, as they did not describe characteristics of the respondent herself. DHS also included multiple variables for the same construct. Age, for example, was captured by multiple variables (continuous age variable and categorical variables with different age categories). Variables were dropped to ensure that each construct was captured by a single variable in the dataset. Once the unnecessary variables were dropped, the researchers identified continuous variables that needed to be categorized, or converted to categorical variables. These categorization decisions were based on how variables were categorized in prior research, to ensure consistency of interpretations with existing literature. Our final analysis included a total of 6561 independent variables.

## Analysis

We used a previously validated approach that includes machine learning models to identify themes (group of variables related to a common topic) associated with an outcome of interest, from a large group of independent variables [23, 24]. The past decade has noted multiple studies that have highlighted the potential of machine learning models in understanding public health related issues. It is a rapidly expanding field, and in its most rudimentary form, machine learning models learn from the data, and identifies patterns or relationships among the variables in the context of prediction, or classification. While there are a variety of machine learning models, the current study uses two specific types of models that are apt for classification tasks (classifying an outcome as pre-defined categories or levels), and have been used in similar prior research: Least Absolute Shrinkage and Selection Operator (lasso) or L-1 regularized regression model, and ridge or L-2 regularized regression model.

**Least Absolute Shrinkage and Selection Operator (lasso).** Lasso is a type of regression model that has been widely used as a powerful tool for data reduction, or feature selection in cases where models have a large number of features or independent variables. [26, 27] The model imposes a penalty on the size of the regression coefficients, trying to shrink them towards zero. [28] The log-likelihood function for lasso takes the form:

$$l_\theta(y|X) = \sum_i -\log(1 + e^{-X_i\theta}) + \sum_{y_i=0} -X_i\theta - \lambda|\theta|$$

Where X is the vector of features or variables and θ is the column vector of the regression coefficients. λ is the tuning parameter, and the term $\lambda|\theta|$ is the regularizer, which allows the model to carry out multiple iterations for the log-likelihood function to find the best values for all the betas (coefficients) in the equation, while mitigating overfitting and bias. The larger the value of λ, the stronger its influence is, and the smaller are the parameter estimates. When λ = 0 the solution is the ordinary maximum likelihood equation. Different approaches to choose the value of λ have been described in existing literature. We use k-fold cross-validation for our lasso model, a method that is described in the later section.

Using the regularizer, the lasso model shrinks the value of coefficients for the features that are least related to the outcome, to an exact zero. For models which are expected to include noise, lasso can thus help identify irrelevant variables by forcing the coefficient values to zero.

**L2 regularized logistic regression model, or ridge.**   Like lasso, ridge is also a type of regularized machine learning model. However, ridge does not force the coefficient values to exactly zero. The log-likelihood function for a ridge model is:

$$l_\theta(y|X) = \sum_i -\log(1 + e^{-X_i\theta}) + \sum_{y_i=0} -X_i\theta - \lambda|\theta|_2^2$$

The tuning parameter, λ, for ridge is also selected using k-fold cross validation.

The order in which the two machine learning models, lasso and L2 regularized models, were implemented in the current study, is described in the following section.

**Preparing dataset for machine learning models.**   As with most machine learning classification models, we first split our dataset to training and test dataset (80:20 ratio- random splitting). The training dataset is where the machine learning models get trained or implemented. Instead of holding back a separate validation dataset, we used k-fold cross validation on the training dataset (with value of k set to 5), to determine the values of the necessary hyperparameters (for example: the tuning parameter, λ, discussed above). In this method, the training dataset is partitioned into 5 subsets of approximately equal size and one of the subsets becomes the validation set. The remaining 4 subsets are used as training data. This procedure is repeated 5 times, each time with a different validation set, and the optimum value of λ is estimated such that the cross-validated log-likelihood is maximized.

We evaluated the performance of these models on the test dataset, by comparing the actual labels (outcomes for each observation) with the labels/outcomes predicted by the machine learning model. We used two evaluation metrics- area under the receiver operating characteristic curve (AUC), and the balanced error rate (BER). The receiver operating characteristic curve is a plot of the test true-positive rate (y-axis) against the corresponding false-positive rate (x-axis); i.e., sensitivity against specificity. AUC provides an estimate of accuracy of our models. BER is the average of true positives and true negatives. For low-prevalence outcomes, or highly imbalanced datasets like ours, AUC and BER provide an accurate estimate of performance of machine learning models.

**Iterative thematic analysis (ITA) with machine learning models.**   As described in Raj et al [23], we used a process of iterative categorization of results from two machine learning models, to identify themes correlated to IPV help-seeking. This process combines quantitative (machine learning models) and qualitative methods. The qualitative efforts include coding of results from statistical machine learning models, into different related and relevant themes. A flowchart depicting the different steps of the process is included in the Supplementary Information files.

According to the ITA process, we first ran a lasso regression model on the training dataset with IPV help-seeking as the outcome, and all eligible variables in the DHS dataset as the independent variables. As noted above, lasso is often used for data reduction; it shrinks coefficient values of irrelevant variables to zero. Since our analysis included a large number of independent variables, our goal with lasso was thus to get rid of the 'noise', or variables completely unrelated to our outcome. Next, we drop all variables with coefficient value zero in the lasso model. We then run a ridge regression model, with the remaining variables as independent factors and IPV help-seeking as the outcome. The results from this ridge regression model constituted the findings from the first round of the ITA process. The coefficient values of all variables were sorted from high to low, and the values were then plotted to identify the point of maximum curvature or the knee point [29] (using *kneed* library in Python). Similar to the prior study using this approach, the variables with coefficient values higher than the knee

point, or the point where the coefficient curve becomes flat, were extracted as the relevant correlates of the outcome [23].

Two researchers reviewed the results from the ridge regression model separately, and coded the variables into different themes. A theme referred to a group of variables that were related to each other in terms of topical similarity. For example, variables *'social class'*, *'religion'*, and *'age'* could be categorized as one theme- socio-demographics. We observed over 95% agreement between the two coders for this process of qualitative thematic categorization (measured as percentage of variables coded as same themes by the coders).

Once the thematic categorization was completed, we proceeded to the next round of the ITA process. We identified the theme which had the maximum variance, i.e., the theme with the variable that has the highest coefficient value. All variables from this theme were dropped from the dataset, and the process of lasso, followed by ridge and qualitative coding was carried out. We continued to repeat this process until no new themes were identified for three consecutive rounds, or no new variables identified for any consecutive round. For each round of ITA, the machine learning models were tested for accuracy and error rates. The resulting output from this process was thus a group of themes or topics that are correlated to IPV help-seeking from anyone.

We repeated the analysis with IPV help-seeking from formal institutions as the outcome.

All analyses were adjusted for sampling weights provided by DHS. The analyses were undertaken in Python with pandas, scipy, keras, numpy, sklearn and tensorflow libraries [code available from authors upon request].

## Results

Fourteen percent of ever-married women who have experienced physical and/or sexual violence in their lifetime reported to have sought help from anyone [Table 1]. Around 9% reached out to their own family for help, and less than 1% sought help from formal institutions (0.6% from police, 0.2% from doctors, 0.1% from social service organizations, and 0.2% from lawyers). Of those who sought help from formal institutions, 61% went to the police. The estimates are not exclusive- one woman could have reported seeking help from multiple sources.

Around half of the sample was literate, with only 12% belonging to households from the highest wealth quintile (richest households). No significant differences were observed for help-seeking from anyone, by any socio-demographic characteristics, except region of residence. However, for help-seeking from formal sources, women differed with regards to education and rural/urban residence.

### Themes associated with seeking help from anyone

We identified 28 variables with coefficient value above the knee point, from the first round of ITA. These 28 variables were coded into six themes: Injury from violence, Controlling behavior/Emotional abuse by husband, History of violence, Alcohol consumption by husband, Health care access and use, and Economic situation.

The theme Injury from violence included variables related to women's experience of wounds, bruises, burns etc. due to the violence perpetrated by their husbands; women experiencing injury were more likely to seek help. Controlling behavior was also positively associated with help-seeking, and it included variables related to emotional violence as well as husband's control over women's daily lives. Similar associations were observed with the theme Alcohol consumption. History of violence covered variables related to women's experience of physical violence during pregnancy, sexual violence experience, perpetration of IPV by woman's father, and woman perpetrating physical violence on their husbands.

**Table 1. Sample characteristics (N = 19,468).**

| | All women who experienced physical or sexual IPV (N = 19,468) | Women who sought help (n = 2,747) (14.4%) | Women who did not seek help (n = 16, 721) (85.6%) | p-value (Chi-square/t-test) | Women who sought help from formal institutions (n = 161) (1.0%) | Women who did not seek help from formal institutions (n = 19, 307) (99.0%) | p-value (Chi-square/t-test) |
|---|---|---|---|---|---|---|---|
| **Characteristics** | **Wtd. %/Mean** | **Wtd. %/Mean** | **Wtd. %/Mean** | | **Wtd. %/Mean** | **Wtd. %/Mean** | |
| Sources of help[1] | | | | | | | |
| Own family | 9.2% | 63.7% | - | | 15.8% | - | |
| Husband/ partner's family | 5.0% | 34.8% | - | | 0.8% | - | |
| Neighbor | 1.7% | 12.0% | - | | 22.0% | - | |
| Friend | 1.9% | 13.2% | - | | 21.5% | - | |
| Social service organization | 0.1% | 0.7% | - | | 12.2% | - | |
| Religious leader | 0.3% | 2.2% | - | | 6.4% | - | |
| Doctor | 0.2% | 1.4% | - | | 22.5% | - | |
| Lawyer | 0.2% | 1.3% | - | | 20.1% | - | |
| Police | 0.6% | 3.8% | - | | 60.9% | - | |
| Other | 0.3% | 1.8% | - | | 1.9% | - | |
| Age | 33.8 | 34.0 | 33.8 | 0.35 | 36.3 | 33.8 | 0.16 |
| Literate | 51.2% | 51.5% | 51.1% | 0.82 | 76.7% | 50.9% | **0.00** |
| Education | | | | | | | |
| None | 43.1% | 43.7% | 42.9% | 0.07 | 22.7% | 43.2% | **0.00** |
| Primary | 17.3% | 16.6% | 17.4% | | 13.9% | 17.3% | |
| Secondary | 35.2% | 33.8% | 35.4% | | 52.5% | 35.0% | |
| Higher | 4.5% | 5.9% | 4.2% | | 10.8% | 4.4% | |
| Household wealth quintile: | | | | | | | |
| Poorest | 24.0% | 25.0% | 23.8% | 0.50 | 16.1% | 24.1% | 0.10 |
| Poorer | 23.8% | 23.6% | 23.9% | | 19.7% | 23.9% | |
| Middle | 21.9% | 20.0% | 22.2% | | 15.9% | 21.9% | |
| Richer | 18.3% | 18.6% | 18.2% | | 24.5% | 18.2% | |
| Richest | 12.0% | 12.8% | 11.9% | | 23.8% | 11.9% | |
| Religion | | | | | | | |
| Muslim | 12.5% | 11.3% | 12.7% | 0.16 | 11.1% | 12.5% | 0.72 |
| Hindu and Others | 87.5% | 88.7% | 87.3% | | 88.9% | 87.5% | |
| Caste | | | | | | | |
| SC/ST | 35.3% | 37.6% | 34.9% | 0.19 | 32.1% | 35.3% | 0.73 |
| OBC | 47.2% | 46.2% | 47.4% | | 52.5% | 47.2% | |
| Other caste/ General | 17.5% | 16.3% | 17.7% | | 15.4% | 17.5% | |
| Place of residence: | | | | | | | |
| Rural | 71.8% | 73.1% | 71.6% | 0.34 | 54.3% | 71.9% | **0.02** |
| Urban | 28.2% | 26.9% | 28.4% | | 45.7% | 28.0% | |
| Region of residence | | | | | | | |

(*Continued*)

**Table 1.** (Continued)

| Characteristics | All women who experienced physical or sexual IPV (N = 19,468) | Women who sought help (n = 2,747) (14.4%) | Women who did not seek help (n = 16, 721) (85.6%) | p-value (Chi-square/t-test) | Women who sought help from formal institutions (n = 161) (1.0%) | Women who did not seek help from formal institutions (n = 19, 307) (99.0%) | p-value (Chi-square/t-test) |
|---|---|---|---|---|---|---|---|
| | Wtd. %/Mean | Wtd. %/Mean | Wtd. %/Mean | | Wtd. %/Mean | Wtd. %/Mean | |
| North | 9.3% | 10.9% | 9.0% | **0.00** | 9.8% | 9.3% | 0.44 |
| West | 10.3% | 8.9% | 10.6% | | 6.4% | 10.4% | |
| South | 26.3% | 29.7% | 25.7% | | 35.6% | 26.2% | |
| Northeast | 2.9% | 1.7% | 3.0% | | 3.1% | 2.8% | |
| East | 26.9% | 23.7% | 27.5% | | 22.4% | 26.9% | |
| Central | 24.3% | 25.1% | 24.2% | | 22.6% | 24.3% | |

[1] Sources of help are not exclusive; women were asked to identify all sources from which they sought help.

Health care access and use included a range of variables on women's use of health services for self and her child, with a specific focus on access to family planning health services. We found that women who knew where to access contraceptives from, and who has been to a health facility were more likely to seek help for IPV. Economic situation covered women's employment status and access to economic resources; income-generating women were more likely to seek help.

The theme Injury from violence had the maximum variance in the first round of ITA. The variables encompassed within this theme were thus dropped in the second round, which identified two new themes: Marital Relationships, and Access to/use of Economic programs [Table 2]. Marital Relationships included marital status (separated or formerly in union), and variables indicating absence of sexual activity in recent months. The theme Access to/use of Economic programs related to women's knowledge of, and non-participation in any self-help groups, or programs that allow women to borrow money to start a business, in their communities. No new variables were identified in the third round of ITA, and hence the iterative process was ended. The accuracy of the machine learning models, as measured by AUC, in the three rounds of ITA was higher than 65% [Fig 1].

### Themes associated with seeking help from formal institutions

Findings for the outcome 'help-seeking for IPV from formal institutions' were similar to the first outcome. Seven themes were identified after four rounds of ITA: Injury from violence, Controlling behavior/Emotional abuse by husband, History of violence, Alcohol consumption by husband, Health care access and use, Economic situation, and Relationships. The themes Injury from violence, Controlling behavior/Emotional abuse by husband, and History of violence included similar variables as were noted for the first outcome, i.e., help-seeking from anyone. Health care access and use did not focus on family planning services as was observed for the previous outcome. This theme included variables indicating woman's agency in accessing health services, as well as her actual use of a health facility for self or for her child. Economic situation focused on women's employment, and the theme Marital relationship is indicative of women being separated from their husband and living with their father/parents [Table 3]. Two additional variables were also identified that could not be categorized into any themes- women's frequent use of television, and source of information for HIV/AIDS. As with the previous outcome, the accuracy of the machine learning models in the four rounds of ITA was higher than 65% [Fig 2].

**Table 2. Themes and their corresponding variables correlated with IPV help-seeking from anyone, based on iterative thematic analysis (ITA).**

| Injury from violence | Controlling behavior/ Emotional abuse by husband | History of violence | Alcohol consumption by husband | Health care access and use | Economic situation | Access to/use of economic programs | Marital Relationship |
|---|---|---|---|---|---|---|---|
| Had bruises because of husband's actions | Woman afraid of husband most of the time | Was physically hurt by someone during pregnancy | Husband drinks alcohol | Has visited health facility for self or child in the last three months | Woman currently working | Woman knows of programs in this area that give loans to women to start or expand a business | Woman's marital status: formerly in union/living with a man |
| Had eye injuries, sprains, dislocations or burns because of husband | Husband jealous if wife talks with other men | Experienced sexual violence first at age 5–18 years | Frequency of husband being drunk: often | Knows of some source to get condoms | Woman does not own a house | Woman has never taken a loan, cash or in kind, from these programs | Time since last sex (in days): 31+ days |
| Had wounds, broken bones, broken teeth or other serious injury because of husband | Husband accuses wife of unfaithfulness | Experienced sexual violence first at age 19–49 years | | Woman usually decides regarding their own health care | Woman does not own land | | Number of sex partners, including spouse, in last 12 months: zero |
| Had had severe burns because of husband | Husband insists on knowing where wife is | Woman physically hurt husband when he was not hurting her | | Knows that private pharmacy is a source for getting condoms | Woman works for a family member | | Reason for not having sex: husband has other women |
| | Husband tries to limit wife's contact with family | Experienced IPV the first time during first year of marriage | | No problem in getting permission for getting medical help for self | Husband's occupation: skilled and unskilled manual work | | |
| | Woman been insulted or made to feel bad by husband/partner | Woman's father beat her mother | | Went for medical treatment for self recently | | | |
| | Husband does not permit wife to meet female friends | | | Woman can get a condom for herself if she wants | | | |
| | Woman been threatened with harm by husband | | | | | | |
| | Husband doesn't trust wife with money | | | | | | |

## Discussion

Despite a high prevalence of IPV in India, only one of every seven women who experienced physical and/or sexual violence from their husbands seek help from anyone, and less than 1% reach out to formal institutions. In line with prior quantitative research from India, we find that experience of severe forms of violence that result in injury, husband's alcohol consumption, and woman's economic independence are some of the key factors influencing a woman's decision in seeking help from anyone, formally or informally [8, 30]. With our exploratory approach involving machine learning models, we identified additional correlates of IPV help-seeking, which are relatively under-studied and have received less focus from existing research efforts.

Study results show that women who experience emotional violence, in addition to physical and/or sexual violence, and multiple forms of controlling behavior by their husband are more likely to seek help from formal and/or informal sources. These findings, combined with the

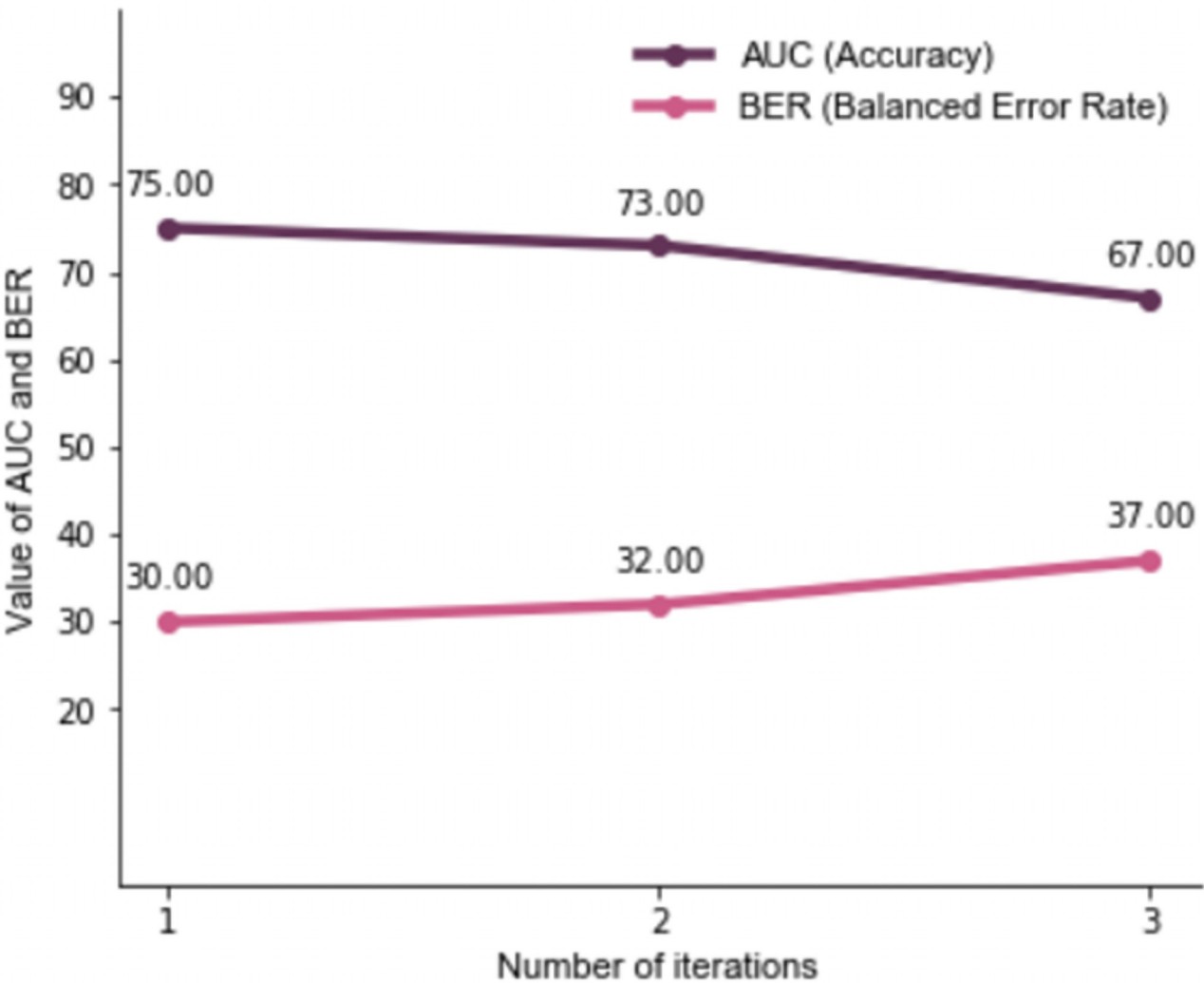

**Fig 1. Accuracy (AUC) and balanced error rates (BER) for models with help-seeking from anyone as outcome.**

observed relationship between injury and help-seeking indicate that women more often look for help only when the violence they are enduring becomes extreme or constant. A study from Bangladesh notes similar findings, with severely abused women in rural areas almost eight times more likely than moderately abused women to seek help [20]. Existing cultural norms in India place the responsibility of maintaining coherence and peace within the family unit on the woman alone. This can often lead to attitudes that justify and accept violent behaviors by husbands, thus discouraging help-seeking [31]. Our findings highlight the need for interventions that include routine IPV screening among married women in India, a country where the social environment prevents most women from disclosing their experiences of violence.

Women who have access to, and use, health services for themselves and their children are also more likely to seek help for IPV. Indicators related to women's ability to make decisions for their own healthcare, and ability to access family planning and other health services are associated with help-seeking behavior. These variables, along with other identified factors related to women's employment, capture the importance of women's agency and autonomy in increasing women's access to help through increased connectivity. Access to health services

**Table 3. Themes and their corresponding variables correlated with IPV help-seeking from formal institutions, based on iterative thematic analysis (ITA).**

| Injury from violence | Controlling behavior/ Emotional abuse by husband | History of violence | Alcohol consumption by husband | Health care access and use | Economic situation | Marital Relationship |
|---|---|---|---|---|---|---|
| Had bruises because of husband's actions | Woman afraid of husband most of the time | Was physically hurt by someone during pregnancy | Husband drinks alcohol | Has visited health facility for self or child in the last three months | Woman currently working | Woman's marital status: formerly in union/living with a man |
| Had eye injuries, sprains, dislocations or burns because of husband | Husband jealous if wife talks with other men | Experienced sexual violence first at age 5–18 years | Frequency of husband being drunk: often | Woman usually decides regarding their own health care | Type of earnings from woman's work: cash only | Relationship to household head: daughter |
| Had wounds, broken bones, broken teeth or other serious injury because of husband | Husband accuses wife of unfaithfulness | Experienced sexual violence first at age 19–49 years | | Woman did not go to a traditional healer for medical help | Husband's occupation: skilled and unskilled manual work | Woman not married and had no sex in last 30 days |
| Had had severe burns because of husband | Husband insists on knowing where wife is | Experienced IPV the first time during first year of marriage | | | | |
| | Husband tries to limit wife's contact with family | | | | | |
| | Woman been insulted or made to feel bad by husband/partner | | | | | |
| | Husband does not permit wife to meet female friends | | | | | |
| | Woman been threatened with harm by husband | | | | | |
| | Husband doesn't trust wife with money | | | | | |

may translate to women's access to certain IPV-related screening or support services in the health facilities. These findings correspond with prior research that document favorable results for women when IPV interventions are integrated with family planning services and economic interventions [32–34]. Unfortunately, however, most health providers in India do not receive training specific to IPV responses during their education [35]. Next, economic independence through employment can provide women with the necessary financial resources to seek help and leave abusive relationships. This is important, given that our analysis also indicates that being separated or being currently unmarried is one of the correlates of IPV help-seeking. It is thus important for formal institutions to take into account necessary rehabilitation of women as a key service.

With regards to correlates of IPV help-seeking from formal institutions, we found that very few women sought help from police, lawyers, doctors or social service organizations in India. This could be due to a lack of knowledge of formal resources for IPV, as well as a lack of access to these sources, fear of stigma, and mistrust that their help-seeking would be acknowledged, validated and respectfully responded to by these formal institutions [36, 37]. Multiple qualitative studies have documented the lack of support received from police by victims of IPV in India [18, 19, 38]. Our findings show that overall, correlates of help-seeking from formal sources are similar to help-seeking from anyone. Although, with regards to help-seeking from formal sources, there is a greater focus on the woman being separated from their husbands, and living with their father/parents. This may be indicative of such services only being accessed

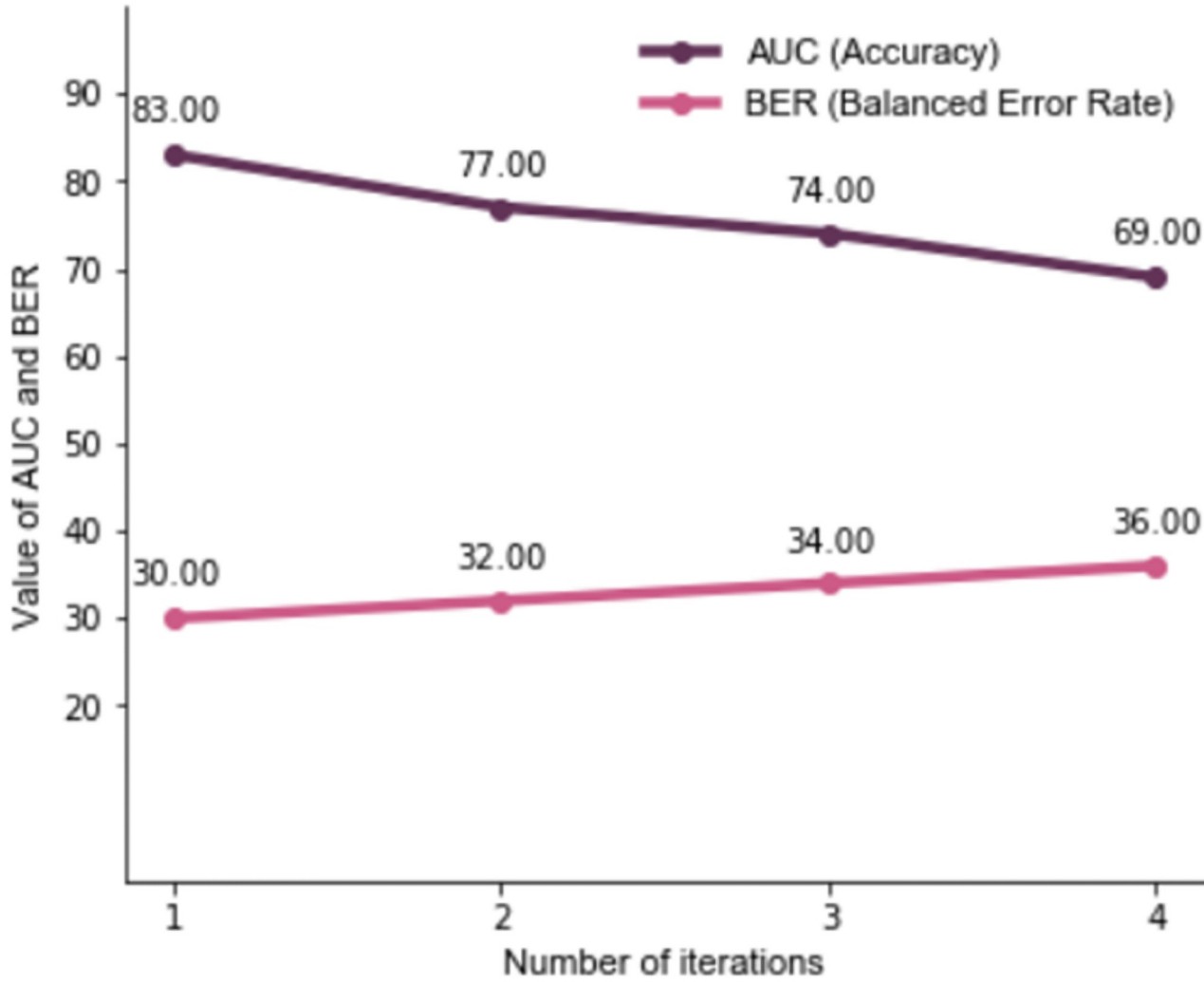

**Fig 2. Accuracy (AUC) and balanced error rates (BER) for models with help-seeking from formal institutions as outcome.**

in cases of separation, or possible separation from the abusive partner. Such findings speak to the need for formal services that can support women who remain with a partner that has been abusive, as this is the case for most women. Such services must include engagement with male partners to stop their abuse. At the same time, given the associations of use of formal services with separation and residence with parents, these findings also highlight the importance of natal families in supporting women affected by IPV.

Our study has a few limitations. First, the survey data used in this study relies on self-report responses and thus is subject to both recall bias and social desirability bias, as well as to the limited generalizability of study findings to India. Second, we used two specific forms of machine learning models. While there are multiple other types of machine learning models that could potentially have better performance than the ones chosen for this study, these two models were selected based on their robust performance in studies with large number of independent variables, as well as their prior use in related studies on gender issues. Next, this analysis is cross-sectional and does not indicate causality. Finally, our approach is exploratory and does

not identify an exhaustive list of correlates for IPV help-seeking. The findings reflect the themes from variables that account for the most variance in our outcome of interest.

## Conclusions

Current study findings are vitally important in characterizing women who are more vulnerable to not disclosing IPV; results highlight the importance of access to social, health, and economic connectivity, particularly in cases of less severe abuse and/or where separation from the abusive partner may be less likely. Our key findings indicate that increased interaction with the health system can raise women's awareness of IPV related services available to them, or increase their access to such services. It may be useful for interventions to consider supporting women more holistically by provision of IPV services integrated with other programs aimed at improving women's health. At the same time, it is important to have community-based interventions to reach women who may be suffering but unwilling to disclose due to internalized gender norms and a lack of economic or social independence.

## Supporting information

**S1 Fig. Flowchart of the iterative thematic analysis process.**
(DOCX)

**S1 Table. Characteristics of all women included in the sample.**
(DOCX)

## Author Contributions

**Conceptualization:** Nabamallika Dehingia, Anita Raj.

**Formal analysis:** Nabamallika Dehingia.

**Funding acquisition:** Anita Raj.

**Methodology:** Lotus McDougal, Julian McAuley, Anita Raj.

**Supervision:** Lotus McDougal, Julian McAuley, Anita Raj.

**Writing – original draft:** Nabamallika Dehingia, Arnab K. Dey.

**Writing – review & editing:** Nabamallika Dehingia, Arnab K. Dey, Lotus McDougal, Julian McAuley, Abhishek Singh, Anita Raj.

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
