## [Decision Letter · Decision Letter 0]

1 Sep 2021

PONE-D-21-21691

Help seeking behavior by women experiencing intimate partner violence in India: a machine learning approach to identifying risk factors

PLOS ONE

Dear Dr. Dehingia,

Thank you for submitting your manuscript to PLOS ONE. After careful consideration, we feel that it has merit but does not fully meet PLOS ONE’s publication criteria as it currently stands. Therefore, we invite you to submit a revised version of the manuscript that addresses the points raised during the review process.

We look forward to receiving your revised manuscript.

Kind regards,

Yukiko Washio, Ph.D.

Academic Editor

PLOS ONE

Journal Requirements:

“This study was funded under a grant from the Bill and Melinda Gates Foundation (Grant number OPP1179208; PI: Anita Raj)”

4. We note you have included a table to which you do not refer in the text of your manuscript. Please ensure that you refer to Table 2 and 3 in your text; if accepted, production will need this reference to link the reader to the Table.

Reviewers' comments:

Reviewer's Responses to Questions

**Comments to the Author**

1. Is the manuscript technically sound, and do the data support the conclusions?

Reviewer #1: Yes

Reviewer #2: Partly

2. Has the statistical analysis been performed appropriately and rigorously? 

Reviewer #1: Yes

Reviewer #2: No

3. Have the authors made all data underlying the findings in their manuscript fully available?

Reviewer #1: Yes

Reviewer #2: No

4. Is the manuscript presented in an intelligible fashion and written in standard English?

Reviewer #1: Yes

Reviewer #2: Yes

5. Review Comments to the Author

Reviewer #1: Overall, the article is clearly written (some minor editing/grammatical errors that can be addressed with an additional review), engages with the literature, and establishes the gap in the literature which it is attempting to address.

Minor Comment: In line 115 the authors indicate that emotional violence is not considered as an outcome. The rational for this could be brought upfront as well rather than only including it in the limitations section.

Reviewer #2: This paper aimed to examine correlates of help-seeking from anyone (e.g. family, friends, police, doctor etc.), as well as help-seeking from any formal sources. To perform analysis, authors used data from a nationally-representative health survey conducted in 2015-16 in India, and included all variables in the dataset (~6000 variables) as independent variables. Two machine learning (ML) models were used- L-1, and L-2 regularized logistic regression models.

Indeed, using ML to identify broad themes associated with help-seeking behavior is a very important technique. If authors can perform the ML appropriately, it would be unique and the results could be very informative.

However, this manuscript doesn’t show the application of ML in the targeted aims.

1) In the “analysis” section, authors only listed some of introduction about ML methods, focusing on LASSO, L-1, L-2, and ITA. Authors didn’t outline the procedure of conducting the analysis.

2) In the “results” section, it is not clear that how the results were obtained from the methods based on the analysis. What’s results were based on LASSO, L-1, L-2 and ITA applied?

3) More importantly, λ and k values were not reported in the results section. AUC and other model measure metrics were not reported or event mentioned.

4) One of the important features based on ML is to rank the importance of the variables after variable selection, but the paper did not even touch the work.

In summary, this paper should be majorly revised

6. PLOS authors have the option to publish the peer review history of their article (what does this mean?). If published, this will include your full peer review and any attached files.

Reviewer #1: No

Reviewer #2: No

---

## [Author Response · Author response to Decision Letter 0]

15 Oct 2021

We thank the reviewers for their relevant and useful comments, which has helped to make the manuscript stronger. Below we respond to each reviewer comment.

Reviewer #1: Overall, the article is clearly written (some minor editing/grammatical errors that can be addressed with an additional review), engages with the literature, and establishes the gap in the literature which it is attempting to address.

Minor Comment: In line 115 the authors indicate that emotional violence is not considered as an outcome. The rational for this could be brought upfront as well rather than only including it in the limitations section.

Thank you for the comment. We have revised the Materials and Methods section to clarify the exclusion of emotional violence from the analysis. Our study does not cover emotional violence as an outcome, given that emotional intimate partner violence (IPV) is often not as agreed upon as physical and sexual IPV, as indicative of abuse and hence requiring help-seeking. Our study focusses on help-seeking for women experiencing sexual and/or physical IPV only- the forms of violence that are recognized more consistently by service organizations, as well as the criminal justice system.

Reviewer #2: This paper aimed to examine correlates of help-seeking from anyone (e.g. family, friends, police, doctor etc.), as well as help-seeking from any formal sources. To perform analysis, authors used data from a nationally-representative health survey conducted in 2015-16 in India, and included all variables in the dataset (~6000 variables) as independent variables. Two machine learning (ML) models were used- L-1, and L-2 regularized logistic regression models.

Indeed, using ML to identify broad themes associated with help-seeking behavior is a very important technique. If authors can perform the ML appropriately, it would be unique and the results could be very informative.

However, this manuscript doesn’t show the application of ML in the targeted aims.

1) In the “analysis” section, authors only listed some of introduction about ML methods, focusing on LASSO, L-1, L-2, and ITA. Authors didn’t outline the procedure of conducting the analysis.

Thank you for this important feedback. We have revised the Materials and Methods section to elaborate the procedure for using the two machine learning models in the study. We have also added a flowchart as supplementary information, that makes the different steps of the analysis clear and easy to interpret. The following text is included in the Analysis section:

"We used a process of iterative categorization of results from two machine learning models, to identify themes correlated to IPV help-seeking. This process combines quantitative (machine learning models) and qualitative methods. The qualitative efforts include coding of results from statistical machine learning models, into different related and relevant themes. A flowchart depicting the different steps of the process is included in the Supplementary Information files.

According to the ITA process, we first ran a lasso regression model on the training dataset with IPV help-seeking as the outcome, and all eligible variables in the DHS dataset as the independent variables. As noted above, lasso is often used for data reduction; it shrinks coefficient values of irrelevant variables to zero. Since our analysis included a large number of independent variables, our goal with lasso was thus to get rid of the 'noise', or variables completely unrelated to our outcome. Next, we drop all variables with coefficient value zero in the lasso model. We then run a ridge regression model, with the remaining variables as independent factors and IPV help-seeking as the outcome. The results from this ridge regression model constituted the findings from the first round of the ITA process. The coefficient values of all variables were sorted from high to low, and the values were then plotted to identify the point of maximum curvature (using kneed library in Python). Similar to the prior study using this approach, the variables with coefficient values higher than the knee point, or the point where the coefficient curve becomes flat, were extracted as the relevant correlates of the outcome. 

Two researchers reviewed the results from the ridge regression model separately, and coded the variables into different themes. A theme referred to a group of variables that were related to each other in terms of topical similarity. For example, variables 'social class', 'religion', and 'age' could be categorized as one theme- socio-demographics. We observed over 95% agreement between the two coders for this process of qualitative thematic categorization (measured as percentage of variables coded as same themes by the coders). Once the thematic categorization was completed, we proceeded to the next round of the ITA process. We identified the theme which had the maximum variance, i.e., the theme with the variable that has the highest coefficient value. All variables from this theme were dropped from the dataset, and the process of lasso, followed by ridge and qualitative coding was carried out. We continued to repeat this process until no new themes were identified for three consecutive rounds, or no new variables identified for any consecutive round. For each round of ITA, the machine learning models were tested for accuracy and error rates. The resulting output from this process was thus a group of themes or topics that are correlated to IPV help-seeking from anyone." 

2) In the “results” section, it is not clear that how the results were obtained from the methods based on the analysis. What’s results were based on LASSO, L-1, L-2 and ITA applied?

Thank you for this comment. The section on the Iterative thematic analysis (ITA) clarifies and elaborates the different steps through which the results were identified. The final results were obtained after an iterative process of running the lasso model, followed by an L2 regression model, and a qualitative coding of the list of variables with coefficient value higher than a specific threshold (the knee point, explained in the prior response). The response to the prior comment includes the description of these steps in detail.

3) More importantly, λ and k values were not reported in the results section. AUC and other model measure metrics were not reported or event mentioned.

Thank you for the comment. We used 5-fold cross validation (k=5) for our analysis. The λ values from the validation for the multiple models of L1/lasso and L2, ranged from 1 to 10^6. The Results section includes a figure that indicates the accuracy estimates, as measured by AUC, and the balanced error rates for each round of the ITA. 

4) One of the important features based on ML is to rank the importance of the variables after variable selection, but the paper did not even touch the work.

Thank you for the comment. In our analysis, we used an iterative approach where we ran lasso/L1 followed by a L2 regression model. The L2 model was run with those variables that had a non-zero coefficient value from lasso- a measure used to get rid of 'noise' or irrelevant variables. The results from the ridge models were then ranked based on the coefficient values. We used the knee point of the coefficient values as the threshold to select relevant variables for the next steps of the ITA process used in our study. The knee point was calculated based on the mathematical definition of curvature for a continuous variable. For any continuous function f, there is a standard closed-form that defines its curvature at any point as a function of its first and second derivative. And, the maximum curvature of this function indicates the levelling off effect. Thus, beyond the knee point, or the point of maximum curvature, the curve for the function becomes flat. We sorted all the coefficient values of the variables from the L2 regression models from high to low. These values were then plotted and assessed to identify the point of maximum curvature or knee point. The variables with coefficient value higher than the knee point were selected as our relevant results. These details have been added to the Materials and Methods section.

---

## [Decision Letter · Decision Letter 1]

28 Dec 2021

Help Seeking Behavior by Women Experiencing Intimate Partner Violence in India: A Machine Learning Approach to Identifying Risk Factors

PONE-D-21-21691R1

Dear Dr. Dehingia,

We’re pleased to inform you that your manuscript has been judged scientifically suitable for publication and will be formally accepted for publication once it meets all outstanding technical requirements.

Kind regards,

Yukiko Washio, Ph.D.

Academic Editor

PLOS ONE

Additional Editor Comments (optional):

Reviewers' comments:

Reviewer's Responses to Questions

**Comments to the Author**

1. If the authors have adequately addressed your comments raised in a previous round of review and you feel that this manuscript is now acceptable for publication, you may indicate that here to bypass the “Comments to the Author” section, enter your conflict of interest statement in the “Confidential to Editor” section, and submit your "Accept" recommendation.

Reviewer #2: All comments have been addressed

Reviewer #3: All comments have been addressed

2. Is the manuscript technically sound, and do the data support the conclusions?

Reviewer #2: Yes

Reviewer #3: Yes

3. Has the statistical analysis been performed appropriately and rigorously? 

Reviewer #2: Yes

Reviewer #3: Yes

4. Have the authors made all data underlying the findings in their manuscript fully available?

Reviewer #2: Yes

Reviewer #3: Yes

5. Is the manuscript presented in an intelligible fashion and written in standard English?

Reviewer #2: Yes

Reviewer #3: Yes

6. Review Comments to the Author

Reviewer #2: The authors have adequately addressed my comments and suggestions raised in a previous round of review. I think that this manuscript is now acceptable for publication.

Reviewer #3: Based on the original reviewers' comments and feedback and the authors' responses, it seems like all comments were adequately addressed. Overall this paper was really interesting to read and presenting an interesting take on machine learning.

7. PLOS authors have the option to publish the peer review history of their article (what does this mean?). If published, this will include your full peer review and any attached files.

Reviewer #2: No

Reviewer #3: No

---

## [Editor Report · Acceptance letter]

24 Jan 2022

PONE-D-21-21691R1 

Help Seeking Behavior by Women Experiencing Intimate Partner Violence in India: A Machine Learning Approach to Identifying Risk Factors 

Dear Dr. Dehingia:

I'm pleased to inform you that your manuscript has been deemed suitable for publication in PLOS ONE. Congratulations! Your manuscript is now with our production department. 

Kind regards, 

on behalf of

Dr. Yukiko Washio 

Academic Editor

PLOS ONE